# Identification of Two Distinct Immune Subtypes in Hepatitis B Virus (HBV)-Associated Hepatocellular Carcinoma (HCC)

**DOI:** 10.3390/cancers16071370

**Published:** 2024-03-30

**Authors:** Davide De Battista, Rylee Yakymi, Evangeline Scheibe, Shinya Sato, Hannah Gerstein, Tovah E. Markowitz, Justin Lack, Roberto Mereu, Cristina Manieli, Fausto Zamboni, Patrizia Farci

**Affiliations:** 1Hepatic Pathogenesis Section, Laboratory of Infectious Diseases, National Institute of Allergy and Infectious Diseases, National Institutes of Health, Bethesda, MD 20892, USA; davide.debattista@nih.gov (D.D.B.); ryakymi@gmail.com (R.Y.); eva.scheibe@gmail.com (E.S.); shinyasato@naramed-u.ac.jp (S.S.); hannahgerstein@gmail.com (H.G.); 2Integrated Data Sciences Section, Research Technologies Branch, National Institute of Allergy and Infectious Diseases, National Institutes of Health, Bethesda, MD 20892, USA; tovah.markowitz@nih.gov; 3NIAID Collaborative Bioinformatics Resource, National Institute of Allergy and Infectious Diseases, National Institutes of Health, Bethesda, MD 20892, USA; justin.lack@nih.gov; 4Department of Surgery, Liver Transplantation Center, Azienda Ospedaliera Brotzu, 09047 Cagliari, Italy; roberto.mereu82@gmail.com (R.M.); faustozamboni@yahoo.it (F.Z.); 5Sevizio di Anatomia Patologica, Azienda Ospedaliera Brotzu, 09047 Cagliari, Italy; cristinamanieli@aob.it

**Keywords:** hepatocellular carcinoma, hepatitis B virus, RNA-sequencing, tumor microenvironment, immune checkpoints in HCC, CTLA-4 expression in HCC

## Abstract

**Simple Summary:**

Chronic hepatitis B virus (HBV) infection remains a major public health problem and the most common risk factor for the development of hepatocellular carcinoma (HCC). The prognosis of HCC is still ominous because diagnosis is usually made at advanced stages and therapeutic options are limited. Immunotherapy is increasingly used for treatment of solid tumors, including advanced HCC. However, most HCC patients do not respond to immunotherapy. The tumor microenvironment (TME) plays a crucial role in intratumor heterogeneity and evolution, treatment failure, and, ultimately, disease outcome. However, there is very limited information on the TME of HBV-HCC. Our study provides evidence that HBV-HCC is characterized by two distinct immune subtypes, immune-high and immune-low. We documented a high expression of CTLA-4 in the immune-high subtype. Our results may have implications in the context of new treatment combinations for HCC to identify patients who might benefit the most from immunotherapy.

**Abstract:**

HBV is the most common risk factor for HCC development, accounting for almost 50% of cases worldwide. Despite significant advances in immunotherapy, there is limited information on the HBV-HCC tumor microenvironment (TME), which may influence the response to checkpoint inhibitors. Here, we characterize the TME in a unique series of liver specimens from HBV-HCC patients to identify who might benefit from immunotherapy. By combining an extensive immunohistochemistry analysis with the transcriptomic profile of paired liver samples (tumor vs. nontumorous tissue) from 12 well-characterized Caucasian patients with HBV-HCC, we identified two distinct tumor subtypes that we defined immune-high and immune-low. The immune-high subtype, seen in half of the patients, is characterized by a high number of infiltrating B and T cells in association with stromal activation and a transcriptomic profile featuring inhibition of antigen presentation and CTL activation. All the immune-high tumors expressed high levels of CTLA-4 and low levels of PD-1, while PD-L1 was present only in four of six cases. In contrast, the immune-low subtype shows significantly lower lymphocyte infiltration and stromal activation. By whole exome sequencing, we documented that four out of six individuals with the immune-low subtype had missense mutations in the CTNNB1 gene, while only one patient had mutations in this gene in the immune-high subtype. Outside the tumor, there were no differences between the two subtypes. This study identifies two distinctive immune subtypes in HBV-associated HCC, regardless of the microenvironment observed in the surrounding nontumorous tissue, providing new insights into pathogenesis. These findings may be instrumental in the identification of patients who might benefit from immunotherapy.

## 1. Introduction

Infection with hepatitis B virus (HBV) remains a major public health problem worldwide. Despite the widespread use of hepatitis B vaccines, in 2019 an estimated 296 million people were chronically infected with HBV, with 1.5 million new infections each year [1]. The dynamic interaction between the virus and the host evolves during the different phases of chronic HBV infection [1]. HBV is a non-cytopathic virus, and it has been established that the liver damage it causes is immune mediated [2]. By immunohistochemistry (IHC), an extensive immune cell infiltration, predominantly in the portal area, has been observed in the immune-active and HBeAg-negative phases of the disease [3]. Interestingly, despite a strong immune response documented in subjects after recovery from acute HBV infection, individuals with chronic hepatitis show impairments in both the innate and adaptive immune responses against HBV [2,4]. The impairment in the adaptive immune response is the result of immune exhaustion, rather than deletion of specific T cells since the adaptive immune response can be restored after viral clearance following antiviral therapy [5]. The long-term sequelae of chronic HBV infection lead to cirrhosis in 30–40% of subjects, whose complications—liver decompensation and hepatocellular carcinoma (HCC)—remain a major global health problem. HCC is the most common form of all primary liver cancers, accounting for approximately 90% of all cases, the 6th most prevalent cancer worldwide and the third leading cause of cancer-related death globally [6,7]. Several environmental and genetic risk factors contribute to the development of HCC, with hepatitis viruses accounting for over 70% of all HCC cases worldwide [8]. Chronic HBV infection is the most common risk factor, accounting for almost 50% of all the cases worldwide [8]. HBV increases the risk of HCC even in the absence of cirrhosis, which remains the most important risk factor for HCC development, being detected in 80–90% of cases [6]. HBV is a DNA virus that can integrate into the host genome, inducing insertional mutagenesis that leads to oncogene activation, such as the *TERT* promoter that causes overexpression of the telomerase reverse transcriptase, which is responsible for the maintenance of telomere length [9]. Moreover, chronic HBV infection causes a state of chronic inflammation, which may lead to immunosuppression, peripheral tolerance, and, ultimately, tumorigenesis [10]. Indeed, it has been shown using cytometry that the tumor microenvironment (TME) of HBV-associated HCC is more immunosuppressive than the TME in non-viral-related HCC [11,12].

The TME in HCC plays a crucial role in tumor heterogeneity and evolution, treatment failure, and, ultimately, patient outcome [13,14]. It is a complex and constantly evolving entity, composed by immune and stromal cells interacting with cancer cells [14]. The generation of an immunosuppressive TME and the impairment of antigen recognition by tumor-specific immune cells lead to immune evasion [15]. Over the past decades, new approaches to immunotherapy have been developed that aim to boost the innate and adaptive anti-tumor immune responses of the host rather than directly targeting cancer cells [16,17,18]. In HCC treatment, the most promising molecules are the immune checkpoint inhibitors (ICIs) [19,20], which are monoclonal antibodies that block the interaction between specific receptors on immune cells and their ligands on tumor cells. Starting in 2011, with the approval of anti-CTLA-4 treatment by the U.S. Food and Drug Administration (FDA) for advanced stage melanoma, several ICIs have been approved for cancer immunotherapy and proven effective in boosting the activation of immune responses against cancer cells [21,22]. However, despite these promising results, around 75% of HCC cases treated with ICIs do not respond to immunotherapy [7,18,19,20]. Among the 25% of subjects with advanced-stage HCC who respond to ICI treatment, it has been documented that viral-related HCC showed an overall higher survival rate compared to non-viral-related HCC [23]. The chronic inflammation that accompanies the development of HCC, which is one of the most heterogenous cancers [24,25,26,27], increases the complexity of the TME [28,29,30], highlighting the need to characterize the TME associated with HCC to better understand the host immune response against cancer cells. Furthermore, most of the studies so far available on the TME have been based on transcriptomic profiles of immune cells in tumor tissues from HCC of different etiologies. Thus, data on the characterization of the immune cell-infiltration and immune checkpoint expression within the tumor and the surrounding nontumorous tissue by immunohistochemistry (IHC), which allows for a more accurate analysis of immune-cell presence and distribution, are lacking.

Taking advantage of a unique series of liver specimens from well-characterized patients with HBV-associated HCC studied at a single center in Italy, we conducted an extensive analysis of the TME in multiple areas of the liver containing HCC by IHC, including CD3, CD8, CD20, alpha-SMA, CD68, CD163, and CD56, and the immune checkpoints CTLA-4 and PD-1 with its ligand PD-L1. We combined IHC analysis with the transcriptomic profile, obtained from RNA-sequencing (RNA-seq), of several areas of the tumor and the surrounding nontumorous tissue to better understand the mechanisms involved in the host immune response within the tumor and to use this knowledge to find new markers that may be used to identify subjects who might benefit from new immune-based therapies.

## 2. Materials and Methods

### 2.1. Patients

We characterized the TME of 12 well-characterized subjects with HBV-associated HCC by studying a total of 42 liver specimens obtained from the tumor and surrounding nontumorous tissue. Each liver sample was divided into two pieces: one was snap-frozen for molecular studies and the other was formalin-fixed and paraffin-embedded (FFPE) for pathological and immunohistochemical analysis. When FFPE sections obtained from the tumor or the perilesional area showed a mixed population of malignant and nonmalignant hepatocytes, the corresponding liver specimen was excluded from molecular studies. One subject was female and 11 were males, with a mean age of 57.7 ± 7.7 years. The clinical, virologic, and histopathological features of these subjects have been previously reported [31]. All were enrolled at the Liver Transplantation Center of the Brotzu Hospital in Cagliari, Italy. Informed consent was obtained from all the subjects involved in this study. The protocol was approved by the Ethical Committee of the Brotzu Hospital (Cagliari, Italy) and by the Office of Human Subjects Research of the NIH, granted on the condition that all samples were deidentified.

### 2.2. Immunohistochemistry

Immunohistochemical staining of formalin-fixed, paraffin-embedded liver sections was performed using a panel of antibodies against CD3, CD8, CD20, CD68, CD56 (Dako), alpha-SMA, PD-1, PD-L1 (Abcam), CD163 (Novus Biologicals), and CTLA-4 (Santa Cruz). In nine patients (five from the immune-high subtype and four from the immune-low subtype), we had the opportunity to analyze four areas: two liver specimens from the tumor, one representing the center (area A) and one the periphery of the tumor (area B); and two from the nontumorous tissue, one from the perilesional area (area C) and one from the most distant area from the center of the tumor (area D, at the edge of the liver). In the remaining three patients (one from the immune-high subtype and two from the immune-low subtype), we stained the center of the tumor (area A) and the most distant area from the center of the tumor (area D, at the edge of the tumor). Briefly, sections of 3 to 5 μm were deparaffinized through graded alcohols and xylene, as previously reported [32,33]. Immunohistochemical staining was performed after antigen retrieval using either citrate buffer (Sigma-Aldrich, St. Louis, MO, USA. 10 mmol, pH 6.0) or EDTA (Sigma-Aldrich, 1 mmol, pH 9.0) with high pressure cooker (115 °C for 10 min). Slides were incubated in blocking solution (PBS supplemented with 0.1% Tween20, 2% bovine serum albumin, and 2% fetal bovine serum) for 1 h at room temperature followed by incubation with streptavidin/biotin blocking kit according to the manufacturer’s recommendations (Vector Laboratories, Newark, CA, USA). The primary antibodies in blocking solution were incubated overnight at 4 °C. Detection was carried out using VECTASTAIN elite ABC HRP kit (anti-mouse or anti-rabbit) according to the manufacturer’s recommendations (Vector Laboratories) and using ImmPACT DAB peroxidase substrate (Vector Laboratories). Hematoxylin solution was used as counterstaining (Sigma). Images were taken using an Olympus BX41 microscope using a digital camera Q-imaging Micropublisher 5.0 RTV. The images were captured using Q-Capture version 3.1.

The presence of tumor-infiltrating lymphocytes (TILs) was evaluated using the expression of CD3 according to the guidelines of the international TIL working group [34]. Briefly, the percentage of TILs were detected in five nonoverlapping areas of the tumor parenchyma and five of the stromal tissue (200× magnification) [34]. TIL density in both areas was measured as percentage of CD3-positive cells. Based on the number of TILs, two tumor subtypes were defined: the immune-low subtype, with a percentage of TILs between 0 and 10%, and the immune-high subtype, with a percentage of TIL greater than 10%. The expression of immune checkpoints (CTLA-4, PD-1, and PD-L1) was estimated over the entire tumor and surrounding nontumorous area. CTLA-4 staining was evaluated by combining both the percentage and intensity of positive cells, as previously described [35,36]. Scores for the percentage of positive cells were: 0 (0%), 1 (<10%), 2 (10–50%), 3 (51–80%), and 4 (>80%); scores for the intensity staining were: 0 (negative), 1 (mild), 2 (moderate), and 3 (intense). The final immunoreactive score (IRS) was obtained by multiplying both percentage and intensity scores; the values were as follow: 0–1 (negative), 2–3 (mild), 4–8 (moderate), and 9–12 (strongly positive). If the CTLA-4 staining was heterogeneous, each intensity of the staining was scored independently and the results were summed, as previously reported [35,36]. The expression of PD-1 was measured using a simple semiquantitative score, as previously described [37]. Briefly, PD-1 staining was graded based on the density of PD-1-positive lymphocytes: 0 (0%), 1 (<1%), 2 (1–10%), 3 (11–50%), 4 (51–90%), and 5 (>90%). The expression of PD-L1 was evaluated using the tumor proportional score (TPS) commonly used in clinical trials [38]. This score is defined as the percentage of viable tumor cells showing partial or complete PD-L1 staining, regardless of the intensity. The specimen was considered positive when the PD-L1 expression had a TPS greater than 1%.

### 2.3. RNA-Seq Data and Statistical Analysis

RNA-seq data from our previous study [33] were reanalyzed according to the two HCC subtypes detected by immunohistochemistry (immune-high and immune-low subtype) (Appendix A). These sequences were previously deposited in the NCBI Sequence Read Archive (SRA) (https://www.ncbi.nlm.nih.gov/sra/; 21 November 2021. accession no. PRJNA719288) [33]. Data were processed using the Pipeliner workflow (https://github.com/CCBR/Pipeliner, 21 November 2021). Reads were trimmed to remove contaminating adapter sequences and low-quality bases using Cutadapt [39] and aligned to the human hg38 reference genome and Gencode release28 using STAR v2.5.2b run in 2-pass mode [40]. RSEM v1.3.0 [41] was used for gene-level expression quantification, and limma v 3.50.3 [42] was used for voom quantile normalization and differential expression analysis. Only genes with more than 0.5 counts per million across at least five samples were used for the downstream analysis. Selection criteria for paired analysis included genes with fold-change greater than +1.0 or lower than −1.0 and FDR-adjusted *p* value less than 0.05. For unpaired analysis, we selected only genes that showed a T-statistic higher than +1.6 or lower than −1.6. Pathway analysis was performed using Ingenuity Pathway Analysis (IPA, version 01-19-00, Qiagen Redwood City, www.qiagen.com/ingenuity, 21 November 2021). The association of genes to pathways was evaluated as the ratio between the number of genes present in the dataset and the total number of genes that belong to the same pathway, and Fisher’s exact test was used to calculate the significance of such association. Fisher’s exact test was also used to compare the frequency of downregulated and upregulated genes observed in our patients. The Mann–Whitney test was used to compare the IHC results between the two tumor subtypes. GraphPad Prism version 10.0.2 for macOS (GraphPad Software, San Diego, CA, USA) was used for graphical representation of the data and statistical analysis. A *p* value of less than 0.05 (2-sided test) indicates statistical significance.

### 2.4. Whole Exome Sequencing

For each patient, DNA obtained from the tumor and the surrounding nontumorous tissue was used for whole exome sequencing (WES). A total of 24 liver samples were examined. The genomic DNA from liver tissue was extracted from frozen liver specimens using the Gentra Puregene Tissue Kit (Qiagen Hildem, North Rhine-Westphalia, Germany) according to the manufacturer’s recommendations. We used the genome-seek (https://github.com/OpenOmics/genome-seek, 21 November 2021) workflow to perform somatic variant calling in tumor/nontumor pairs. In brief, the fastp v0.23.2 [43] was used to trim reads which were then mapped to the GRCh37 human reference genome with bwa-mem2, and deduplicated BAM files underwent Indel realignment with GATK v3.8 and quality recalibration using GATK4 v4.4.0.0. Recalibrated BAM files were then used to perform paired variant calling with the non-edited naïve samples serving as the control using four variant callers: Octopus v0.7.4 [44], Strelka2 [45], Muse [46], and MuTect2. Variants were initially filtered to include only those called by at least 2 variant callers, variants with ≥3 reads supporting the mutant allele, and variants rare in the general population (<0.001 in Gnomad v3 and 1000 Genomes). All somatic variants were then annotated with VEP v106 and converted to MAF file format using vfc2maf v1.6.21. The WES data have been deposited in the NCBI Sequence Read Archive (SRA) reference number PRJNA1085312 (https://www.ncbi.nlm.nih.gov/sra/, 21 November 2021).

## 3. Results

### 3.1. Characteristics of the Patients

We studied 12 patients with HBV-associated HCC whose clinical, virologic, and histopathological features were previously reported [31]. All patients were HBsAg positive, HBeAg negative, and positive for antibodies to HBcAg (anti-HBc) and to HBeAg (anti-HBe), with very low levels of HBV DNA replication (mean ± SD, 1.8 ± 1.3). All were under nucleos(t)ide therapy. The tumor size was between 2 and 3 cm in eight patients and larger than 3 cm in the remaining four. The grade of tumor differentiation was G2 in 10 patients, G3 in one, and G4 in the remaining patient [31].

### 3.2. Identification of Two Distinct Immune Subtypes in HBV-HCC by Immunohistochemistry

We performed an extensive immunohistochemical analysis to characterize the levels and features of immune-cell infiltration both in the tumor and in the surrounding nontumorous tissue. In nine out of 12 patients, we analyzed several compartments of the liver, including two liver specimens from the tumor, one representing the center (area A) and one the periphery of the tumor (area B); and two from the nontumorous tissue, one from the perilesional area (area C) and one from the most distant area from the center of the tumor (area D, at the edge of the liver). In the remaining three patients, we examined two liver specimens, one from the tumor and one from the surrounding nontumorous tissue. The panel of immunological markers included: CD3 (pan-T cells), CD8 (CD8 T cells), CD20 (B cells), alpha-SMA (a marker of hepatic stellate cell activation), CD68 (monocytes/macrophages), CD163 (M2-like macrophages), and CD56 (NK cells).

Our analysis of CD3-positive cell infiltration in the center of the tumor (area A) provided evidence that HBV-HCC exhibits two distinct subtypes. Half of the patients (six out of 12) that we defined as immune-high subtype (Figure 1) showed a rich infiltration of immune cells, while the remaining 50% of the patients, who we defined as immune-low subtype (Figure 2), showed absence or minimal presence of immune cells [percentage of TIL infiltration (mean ± SEM): immune-high, 26.6 ± 4.9 vs. immune-low, 1.5 ± 0.6; Mann–Whitney test *p* = 0.0022; Appendix A]. The immune-high subtype was characterized by abundant CD3-positive T cell infiltration, usually present as clusters of different sizes, with about 50% of these cells expressing CD8 (Figure 1). We also observed an extensive CD20-positive B cell infiltration within the tumor, although in all cases but one, the number of B cells was less than that of T cells (Figure 1). Most of the immune cells in the immune-high subtype clustered together to form tertiary lymphoid tissue-like structures within the tumor stroma as shown in one representative case in Figure 3. However, in a few cases, along with the clusters, we also observed a high number of single immune cells scattered within the tumor parenchyma. All patients with the immune-high subtype exhibited more than three immune clusters within the tumor, which were characterized by CD20-positive B cells surrounded by a high number of CD3-positive T cells, and rare cells positive for CD68 and/or CD163 (Figure 3). The immune-high subtype was also distinguished by a high number of alpha-SMA positive cells (Figure 1), indicating stromal activation in these tumors. In contrast to the immune-high subtype, the immune-low subtype was characterized by a low number of infiltrating immune cells, which were distributed as single cells within the tumor (Figure 2) without evidence of cluster formation, except in one patient (Pt. C2), who showed a single small cluster of cells positive for CD3, CD8, and CD20. This subtype, in contrast to the immune-high, was associated with a low number of alpha-SMA positive cells (Figure 2), indicating a lower stromal activation in these tumors. In the nine patients from whom four liver specimens were analyzed, we found that the pattern observed in the center of the tumor was also seen at the periphery of the tumor in both subtypes (Appendix A). Remarkably, despite the detection of two distinct subtypes in the tumor, the surrounding nontumorous tissue showed a similar pattern in both the immune-high (Figure 4) and the immune-low subtypes (Figure 5), with extensive immune-cell infiltration (CD3, CD8, and CD20) both in the perilesional area and at the edge of the liver, as shown in a representative case for each subtype (Appendix A).

Next, we analyzed the presence of CD68 and CD163 (expressed by monocytes and Kupffer cells [47]) and CD56 (expressed by NK cells) within the tumor and the surrounding nontumorous tissue. Interestingly, we found that CD68 and CD163 did not differentiate the two subtypes, with a similar number of positive cells in both the tumor parenchyma (Appendix A) and the surrounding nontumorous tissue (Appendix A). Since CD163 is a marker of M2-like macrophages, these results suggest that the infiltrating macrophages belong to the M2-like subtype. In both the tumor and the surrounding nontumorous tissue, we observed only rare, single CD56 positive (Appendix A).

### 3.3. Expression of Immune Checkpoint Molecules in Immune-High and Immune-Low Subtypes by Immunohistochemistry

To better characterize the TME, we investigated by IHC the expression of three immune checkpoint molecules (CTLA-4, PD-1, and PD-L1), which are the targets of ICI therapies approved by the FDA [18], in different compartments of the liver in the immune-high and immune-low HCC subtypes. The most prevalent and abundant immune checkpoint molecule observed in our HBV-HCC cases, regardless of the immune phenotype, was CTLA-4. It was detected in all six immune-high HCC subtype with a strong cytoplasmic staining of cancer cells (Figure 6A) and in four of six immune-low HCC subtype with a less intense staining (Figure 6B). Notably, CTLA4 was detected in the surrounding nontumorous tissue of all patients, regardless of the immune subtype (Figure 6C,D). The distribution of CTLA-4 staining was not homogeneous in all positive liver samples, so we used the IRS to compare the levels of CTLA-4 expression in the two subtypes. Analysis of the IRS within the tumor showed that the immune-high subtype was characterized by significantly higher levels of CTLA-4 expression in cancer cells compared to the immune-low subtype (Mann–Whitney test *p* < 0.0001; Appendix A). Both subtypes showed a similar CTLA-4 expression pattern in the nontumorous tissue (Appendix A).

Regarding PD-1, we found that it was expressed in all immune-high tumors in the infiltrating lymphoid cells, mostly within the immune-cell clusters (Figure 7A). Among these tumors, two showed a high number of PD-1-expressing immune cells (score: 2–3), while the remaining four cases showed rare PD-1-positive cells (score: 1). In the immune-low subtype, we did not observe PD-1-positive cells, except in one case, which showed a very low number of isolated PD-1-positive cells in the tumor parenchyma (score: 1) (Figure 7B). Outside the tumor, very low numbers of PD-1-positive cells were detected in three cases from both subtypes (Figure 7C,D). In all PD-1-positive liver tumors, we found that only a small proportion of CD8 T cells expressed PD-1 (Figure 8). Semiquantitative analysis of PD-1 staining within the tumor showed that the immune-high subtype had a significantly higher frequency of PD-1-positive cells compared to the immune low-low subtype (Mann–Whitney test *p* = 0.0009; Appendix A), while there was no difference between the two subtypes in the surrounding nontumorous tissue (Appendix A).

Next, we analyzed the expression of the PD-1 ligand, PD-L1, which was very weak in both the tumor and the surrounding nontumorous tissue in both HCC subtypes (Figure 9 and Appendix A). In the immune-high subtype, we identified four patients positive for PD-L1 (Figure 9A), defined as >1% of positive cancer cells, as previously reported [38], while the immune-low subtype showed only two patients positive for PD-L1 (Figure 9B). Outside the tumor, both subtypes showed three cases positive for PD-L1 (Figure 9C,D). Interestingly, only two patients in the immune-high subtype were positive for both PD-1 and PD-L1 within the tumor (Pt. H3 and Pt. H4), but the positive cells were localized in different areas of the tumor.

### 3.4. Gene Expression Profiles in Immune-High and Immune-Low Subtypes of HBV-Associated HCC

To better understand the differences between immune-high and immune-low subtypes in HBV-HCC patients, we analyzed RNA-seq data according to the two HCC subtypes, including five patients with the immune-high and five with the immune-low subtype. As a first step, we analyzed the genes differentially expressed between tumor and nontumorous tissue within each subtype. In the immune-high subtype (Figure 10A), we detected 1810 differentially expressed genes within the tumor, with 54.5% downregulated (986 genes) and 45.5% upregulated (824 genes). Analysis of the 20 top-scored pathways in the immune-high subtype (Figure 10B) documented two pathways related to cell cycle regulation, namely kinetochore metaphase signaling pathway and mitotic roles of Polo-like kinase, containing most upregulated genes (85.4%). Among these genes, we found upregulation not only of kinases (e.g., *AURKB*, *BUB1*, *BUB1B*, *CCNB1*, *CDK1*, *NEK2*, and *TTK*) and phosphatases (e.g., *PPP1R14B* and *PPP2R3B*), but also of proteins involved in cell cycle and centromere formation (e.g., *CCNB2*, *CDC20*, *CDCA8*, *CENPA*, *CENPE*, *CENPQ*, *CENPU*, *CENPW*, *H2AC18/H2AC19*, *H2AZ1*, and *PTTG1*). We also identified additional upregulated genes not included in these two molecular pathways, but functionally related to cell cycle regulation (e.g., *CCNA2*, *CDK4*, *E2F7*, *E2F8*, *HDAC11*, *HDAC6*, *MKI67*, and *TGFB3*). Interestingly, all these genes were differentially expressed only in the immune-high subtype. In addition, in this subtype we found 12 pathways related to cell signaling in response to extracellular stimuli (e.g., STAT3 pathway, growth hormone signaling, IL-6 signaling, acute phase response signaling, and JAK/STAT signaling), with most genes downregulated (Figure 10B; 68.8%). Within these pathways, we found downregulation not only of transmembrane receptors (e.g., *EGFR*, *ESR1*, *GHR*, *IFNAR1*, *IFNLR1*, *IL18R1*, *IL2RB*, and *IL4R*) and cytoplasmatic enzymes (e.g., *HMOX1*, *JAK2*, *MAP2K1*, *MAP2K3*, *PIK3R1*, *PPP2CB*, *PPP2R2A*, and *SOCS3*), but also of secreted proteins (e.g., *IGF1*, *IGF2*, *IGFALS*, *IGFBP3*, *IL1RN*, *IL33*, *KLKB1*, *ORM1*, *ORM2*, and *TGFA*). Among the upregulated genes, we found genes related to cell cycle and transcription factors (e.g., *CDK4*, *CDKN2A*, *CFL1*, *E2F7*, *E2F8*, *HDAC11*, *PIAS3*, *PPP2R3B*, and *WASF1*). The remaining pathways were related to cell metabolism (CLEAR signaling pathway, PPARa/RXRa activation, autophagy, and LXR/RXR activation), cell migration (axonal guidance signaling), and oxidative stress and apoptosis (ferroptosis signaling pathway).

In the immune-low subtype (Figure 10C), we detected only 436 differentially expressed genes with a higher proportion of downregulated genes (354; 81.2%) compared to the immune-high subtype (Fisher’s exact test *p* < 0.0001). The remaining genes were upregulated (82; 18.8%). In the immune-low subtype, analysis of the 20 top-scored pathways showed that most of the downregulated genes were involved in pathways related to the metabolism of various substrates (chondroitin sulfate, dermatan sulfate, vitamin A, estrogen, heparan sulfate, melatonin, tryptophan, and thioredoxin) (Figure 10D). In contrast to the immune-high subtype, we identified only two pathways related to cell signaling (osteoarthritis pathway and growth hormone signaling), the latter also downregulated in the immune-high phenotype. Finally, we found several downregulated genes related to phagosome formation and ID1 signaling pathways. Together, these results indicate that the immune-high HCC subtype was specifically characterized by activation of cell proliferation and reduced activation of immune cells compared to the nontumorous tissue, while the immune-low HCC subtype was characterized by a metabolism switch-off. A complete list of differentially expressed genes in the immune-high and immune-low subtypes is reported in Appendix A, respectively.

Next, to better understand the differences between the immune-high and immune-low subtypes, we directly compared the gene expression profiles of the two subtypes. Pathway analysis showed that the immune-high subtype was characterized by inhibition (negative Z-score) of six pathways associated with cytokine production and cell-mediated immune response (neuroinflammation signaling pathway, PD-1, PD-L1 cancer immunotherapy pathway, TREM1 signaling, pathogen-induced cytokine storm signaling pathway, role of pattern recognition receptors in recognition of bacteria and viruses, and Th1 pathway) (Figure 10E and Appendix A). Interestingly, we also observed two activated pathways with a Z-score higher than 1 (WNT/Ca+ pathway, IL-10 signaling). The WNT/Ca2+ signaling seems to have a role in cancer stem cell renewal and proliferation [48], suggesting that the immune-high subtype is characterized by a higher number of genes involved in proliferation than the immune-low subtype. In addition, we observed activation of the IL-10 signaling pathway, which indicates that the immune-high subtype is characterized by an immune-suppressive microenvironment. Finally, we observed an enrichment of downregulated genes associated with antigen presentation, especially associated with MHC class II. Together, these results suggest that, despite the abundant T and B cell infiltration in the immune-high subtype, these immune cells may be non-functional and may not actively defend against the tumor.

To investigate whether the two distinct phenotypes were associated with specific mutational signatures, we also performed WES to study the mutational landscape of our HCC samples, and the data were analyzed according to the two subgroups. As shown in Figure 11, this analysis documented that four out of six individuals with the immune-low subtype were characterized by missense mutations in the CTNNB1 gene, while only one patient had mutations in this gene in the immune-high subtype. Interestingly, mutation of this gene has previously been associated with the immune-low subtype in melanoma, where activation of the WNT/beta-catenin pathway was associated with the immune-cell exclusion phenotype [49].

## 4. Discussion

HCC remains one of the most important public health problems worldwide. To date, the prognosis of HCC is still ominous because the available therapeutic options for advanced-stage disease are very limited and diagnosis is usually made at advanced stages. The mean survival of patients treated with systemic agents has improved by a few months since the introduction of sorafenib in 2008, which has remained the only available therapy for advanced HCC for a decade [50]. One of the most important achievements in cancer therapy has been the emergence of immunotherapy, which is based on the principle of boosting the innate and adaptive anti-tumor immune response of the host rather than directly targeting cancer cells, as done by tyrosine kinase inhibitors such as sorafenib. Since the introduction of ipilimumab (an anti-CTLA-4 antibody) in 2010, ICI therapies have become widely used for the treatment of patients with solid tumors and, more recently, with advanced-stage HCC [18,19,20]. However, despite its increasing use, not all patients with HCC respond to immunotherapy. Only one combination treatment in advanced HCC, atezolizumab (anti-PD-L1) plus bevacizumab (anti-VEGF), has received full FDA approval as a first-line therapy in HCC, while three regimens, nivolumab and pembrolizumab (both anti-PD-1) monotherapy and nivolumab with ipilimumab, have received accelerated approvals [18]. The evidence so far accumulated indicates that only 15% of HCC patients treated with ICI monotherapy, regardless of the tumor etiology, show an effective response to treatment, as evaluated from non-invasive imaging results (e.g., tumor diameter reduction or disappearance and number of lesions), while the combination of two ICIs, or the combination of one ICI with a tyrosine kinase inhibitor, increases the overall response rate to 25% [20,51]. Notably, two meta-analysis studies that analyzed three phase III studies of ICI treatment (CheckMate459, IMbrave150, and KEYNOTE-240) reported a significantly better outcome [52] and a higher overall survival [23] in viral-related HCC than in HCC associated with non-viral etiology. Nevertheless, no predictive biomarkers have been identified and the molecular mechanisms of response and resistance remain poorly understood. Moreover, there is limited information on the characterization of the TME according to the different viral etiology. Thus, attempts to identify and stratify patients that may have a better response to immunotherapy remain a high priority, especially in HCC [53]. Recently, two studies suggested a correlation between immune-cell infiltration and response to ICI treatment [54,55], although a predictive role of PD-1 and PD-L1 expression in the tumor, as observed in other tumors, remains elusive in HCC [38].

In the present study, we characterized the TME of HBV-associated HCC by combining IHC with RNA-seq in paired liver specimens from the tumor and matched nontumorous tissue obtained from several areas of the liver. Our aim was to better understand the mechanisms involved in the host immune response against the tumor and to use this knowledge to identify patients who might benefit from immunotherapeutic approaches. Notably, most of the previous reports analyzed the HCC TME using only the transcriptomic profile, which documented that about 25% of the cases displayed a higher expression of immune-related genes compared to the others [56,57,58]. To date, only one study has used IHC to investigate the TME, but without a clear characterization of the tumor etiology [59]. Another study used multiparametric flow cytometry to show that the tumor microenvironment of HBV-associated HCC is more immunosuppressive than that in non-viral-related HCC [11]. Taking advantage of a series of paired liver specimens, in this study we had the unique opportunity to compare the TME with the surrounding nontumorous tissue. Our extensive IHC analysis provides evidence that HBV-HCC is characterized by two distinct subtypes: immune-high and immune-low. Half of the studied patients belonged to the immune-high subtype, characterized by abundant T and B cell infiltration. These cells were typically organized in clusters forming tertiary lymphoid tissue structures within the stroma of the tumor, localized close to the tumor parenchyma. One of the peculiar features of the immune-high HCC subtype was a strong activation of the tumor stroma, as shown by the positivity for alpha-SMA. Cancer-associated fibroblasts (CAF) constitute the main component of the tumor stroma and are closely associated with tumor infiltration, progression, stemness, chemoresistance, and prognosis [60]. Alpha-SMA is expressed by multiple CAF subsets and is commonly used as marker of CAF activation in liver cancer [61]. Many studies have shown that interaction between CAF and immune cells, as well as other immune components, may modulate the tumor microenvironment and, in some cases, inhibit the anti-tumor immune response [62]. However, further studies are needed to better understand the relationship between CAF activation and tumor immune-cell infiltration in HCC. The availability of paired nontumorous tissue gave us the opportunity to analyze the nontumorous tissue and demonstrate that all patients, regardless of the tumor subtypes, present an immune-high profile in the surrounding nontumorous tissue with a high number of immune cell clusters, containing CD3, CD8, and CD20 positive cells, localized mostly in the periportal space, along with a strong activation of hepatic stellate cells. These results are in line with previous reports based on transcriptomic analysis [57,63], and support the hypothesis that the TME is independent from the surrounding nontumorous tissue, which suggests that HCC can implement mechanisms to exclude immune cells from the tumor, as seen in the immune-low HCC subtype.

The expression of immune checkpoint receptors represents an important mechanism of immune modulation by the tumor. Our study showed that all six immune-high tumors were positive for PD-1, although only two cases had high numbers of positive cells and not all infiltrating CD8 T cells were positive. PD-L1 was observed in four out of six patients with the immune-high subtype, but only two patients expressed both PD-1 and PD-L1, and not in the same area of the tumor. Conversely, only one patient with the immune-low subtype showed rare immune cells positive for PD-1 dispersed in the tumor parenchyma, and only two patients were positive for PD-L1. Notably, the association between PD-1 and PD-L1 expression within the tumor and response to ICI treatment in HCC remains elusive [38]. Further prospective studies are needed to prove the association among PD-1 and PD-L1 expression, immune-high subtype, and ICI treatment efficacy in patients with HBV-HCC.

Recently, the FDA granted accelerated approval to the anti-PD-1 antibody nivolumab in combination with the anti-CTLA-4 ipilimumab for the treatment of patients with HCC previously treated with sorafenib [18]. To the best of our knowledge, this is the first study that has extensively analyzed the expression of CTLA-4 in several areas of the tumor and nontumorous tissue from the same patients with HBV-associated HCC using IHC. We observed a high expression of CTLA-4 in tumor cells in all patients with the immune-high subtype and in four of the six cases with immune-low subtype, although at significantly lower levels than in the immune-high subtype. These results may have important implications in the context of new treatment combinations in HCC patients, even though a complete elucidation of the CTLA-4 biology is needed to understand the function of this molecule on cancer cells. The first study that documented CTLA-4 expression in tumor cells, published in 1997 in human lymphomas [64], suggested that CTLA-4 promotes tumor escape by inhibiting anti-tumor responses but not the proliferation of malignant lymphocytes [64]. Recently, three studies on lung cancer showed that increased tumor expression of CTLA-4 was associated with better outcomes after tumor resection [35,65,66]. However, in some breast cancers [67,68], thymomas [69], esophageal carcinomas [70], and nasopharyngeal carcinomas [71], expression of CTLA-4 correlated with poor prognosis.

In this study, we investigated the TME by combining IHC with RNA seq. Transcriptomic analysis of the tumor tissue showed a predominance of upregulated genes related to cell cycle regulation in the immune-high subtype, suggesting a more pronounced cell proliferation in tumors showing a high immune infiltration compared to the surrounding nontumorous tissue. However, it is not clear if this feature is related to cancer cell proliferation or immune cell proliferation. Further studies are needed to better understand this observation. Conversely, the immune-low subtype was mainly characterized by a molecular profile associated with the downregulation of genes related to cell metabolism, as previously observed by our group using microarray [31]. Interestingly, direct comparison of the two subtypes showed an enrichment of downregulated genes associated with Th1 pathway and antigen presentation, indicating that despite a high number of infiltrating T cells in the immune-high subtype these cells are not active. In addition, downregulation of genes related to the Th1 pathway and antigen presentation was observed in association with the higher expression of CTLA-4, supporting the hypothesis that CTLA-4 may be associated with an impairment of antigen presentation, leading to an inhibition of Th1 and CTL activation, as previously reported in breast cancer [68].

## 5. Conclusions

Our study provides evidence that HBV-associated HCC is characterized by two distinct immune subtypes, immune-high and immune-low, regardless of the high numbers of immune-cell infiltration consistently observed in the surrounding nontumorous tissue. The immune-high subtype was characterized by a higher level of B and T cell infiltration associated with stromal activation, CTLA-4 expression, and a transcriptomic profile characterized by a high proportion of upregulated genes mainly associated with cell cycle. Conversely, the immune-low subtype was characterized by a reduced immune-cell infiltration, a high proportion of downregulated genes associated with cell metabolism, and missense mutations in the CTNNB1 gene in the vast majority of patients. The high expression of CTLA-4 observed in the immune-high subtype seems to be associated with an impairment in antigen presentation and Th1 and CTL responses, as shown by RNA-seq analysis. These patients might be those who benefit the most from immunotherapy targeting CTLA-4, as seen in other tumors. The limit of our study is the relatively low number of patients studied, although the patients were well-characterized and all enrolled at the same center in Italy. Large prospective studies are needed to investigate the clinical significance of the two HBV-HCC immune subtypes and to evaluate if the immune profile may help to identify patients who are the best candidates for immunotherapy.

## Figures and Tables

**Figure 1 cancers-16-01370-f001:**
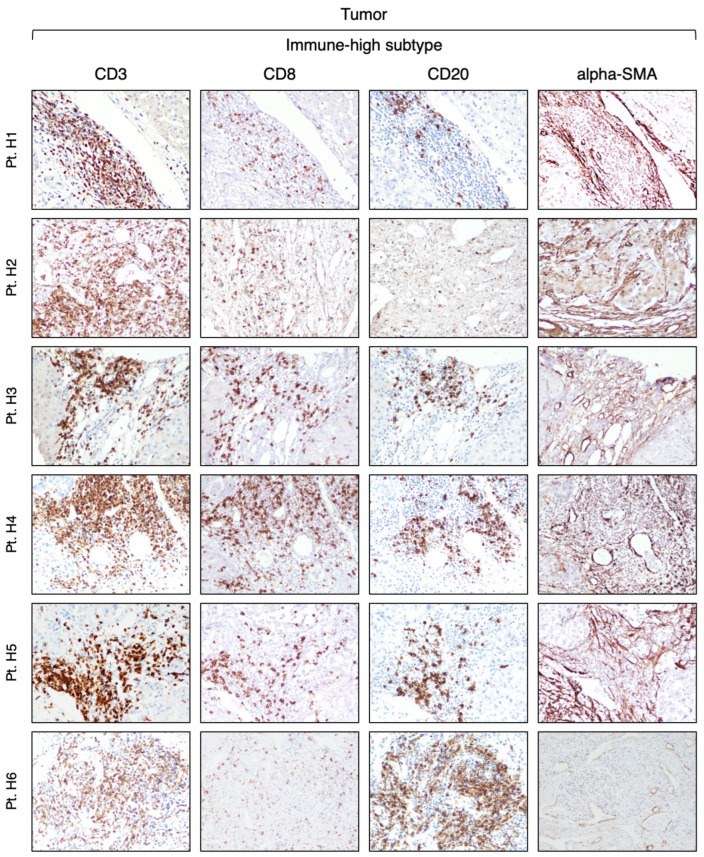
Expression of immune markers within the tumor of the immune-high HCC subtype. The images illustrate the immunostaining with monoclonal antibodies against CD3 (T cells), CD8 (CD8 T cells), CD20 (B cells), and alpha-SMA (stromal activation) in paraffin liver sections from six HBV-HCC patients with the immune-high subtype taken at the time of liver transplantation from the center of the tumor (area A) (200× magnification).

**Figure 2 cancers-16-01370-f002:**
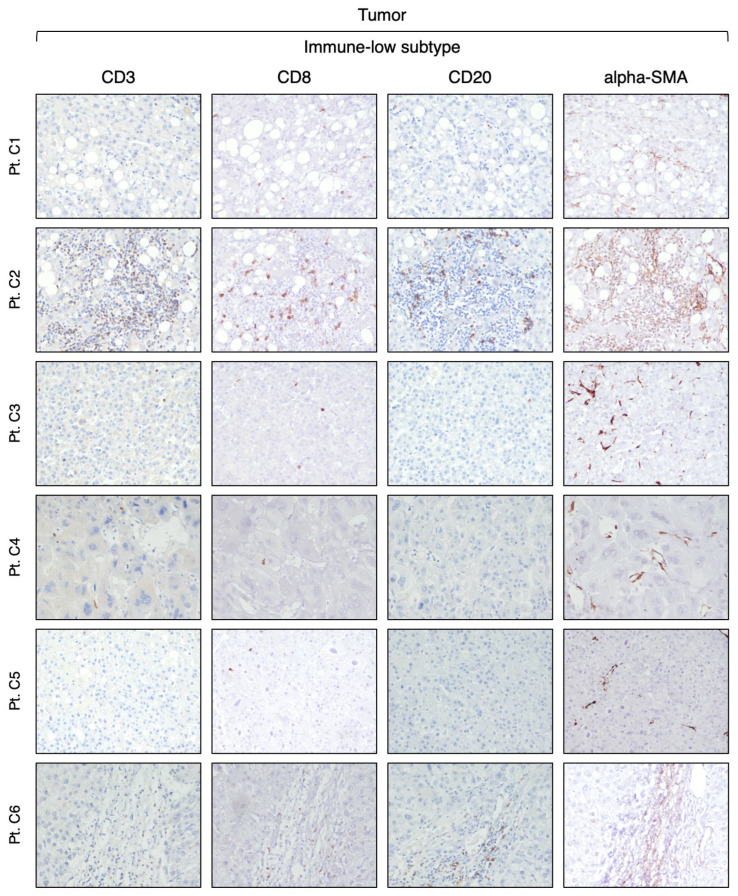
Expression of immune markers within the tumor of the immune-low HCC subtype. The images illustrate the immunostaining with monoclonal antibodies against CD3 (T cells), CD8 (CD8 T cells), CD20 (B cells), and alpha-SMA (stromal activation) in paraffin liver sections from six HBV-HCC patients with the immune-low subtype taken at the time of liver transplantation from the center of the tumor (area A) (200× magnification).

**Figure 3 cancers-16-01370-f003:**
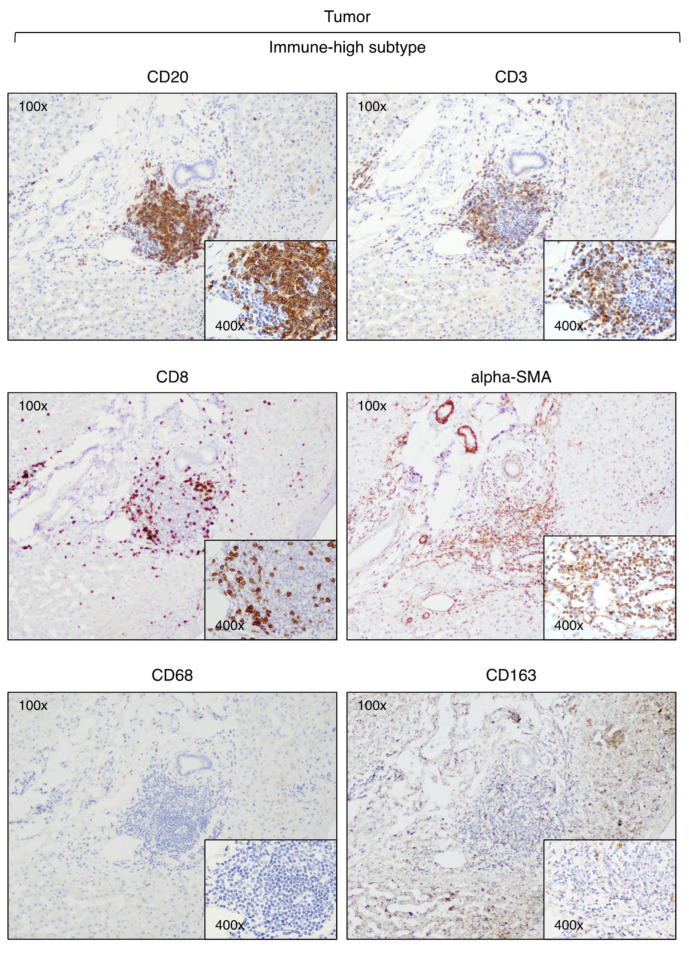
Expression of immune markers within tertiary lymphoid tissue-like structures detected in the tumor of a representative HCC case of the immune-high subtype (Pt. H5). The images illustrate immunostaining of paraffin liver sections taken at the time of liver transplantation from the center of the tumor with monoclonal antibodies against CD3 (T cells), CD8 (CD8 T cells), CD20 (B cells), alpha-SMA (stromal activation), CD68 (monocytes and Kupffer cells), and CD163 (M2-like macrophages) (100× and 400× magnification).

**Figure 4 cancers-16-01370-f004:**
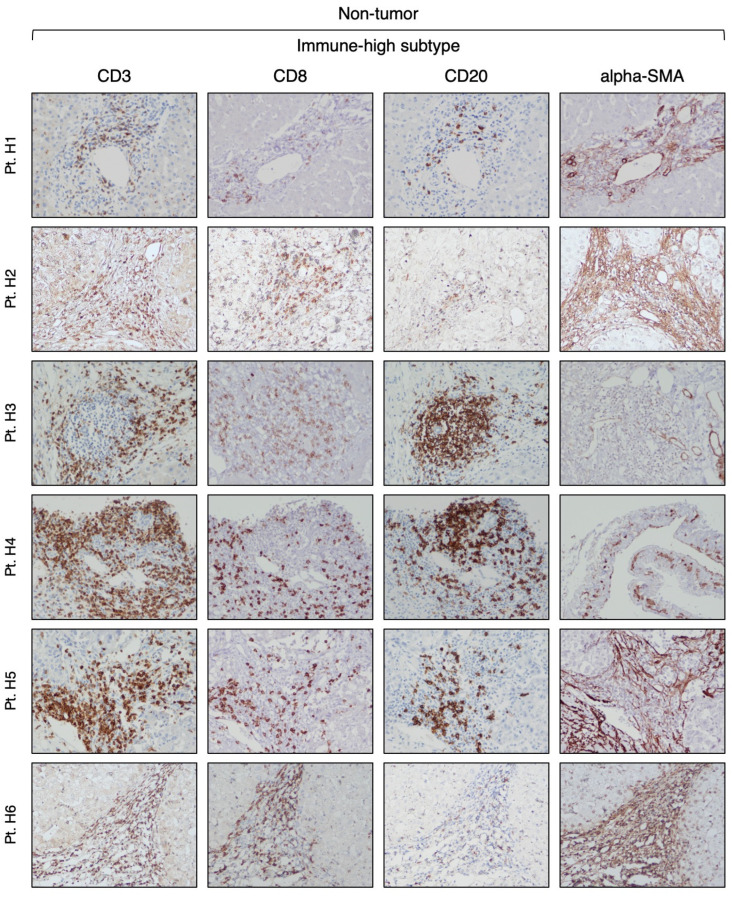
Expression of immune markers in the surrounding nontumorous tissue of the immune-high HCC subtype. The images illustrate the immunostaining with monoclonal antibodies against CD3 (T cells), CD8 (CD8 T cells), CD20 (B cells), and alpha-SMA (hepatic stellate cells) in paraffin liver sections from six HBV-HCC patients with the immune-high subtype taken at the time of liver transplantation from the most distant area from the center of the tumor (area D) (200× magnification).

**Figure 5 cancers-16-01370-f005:**
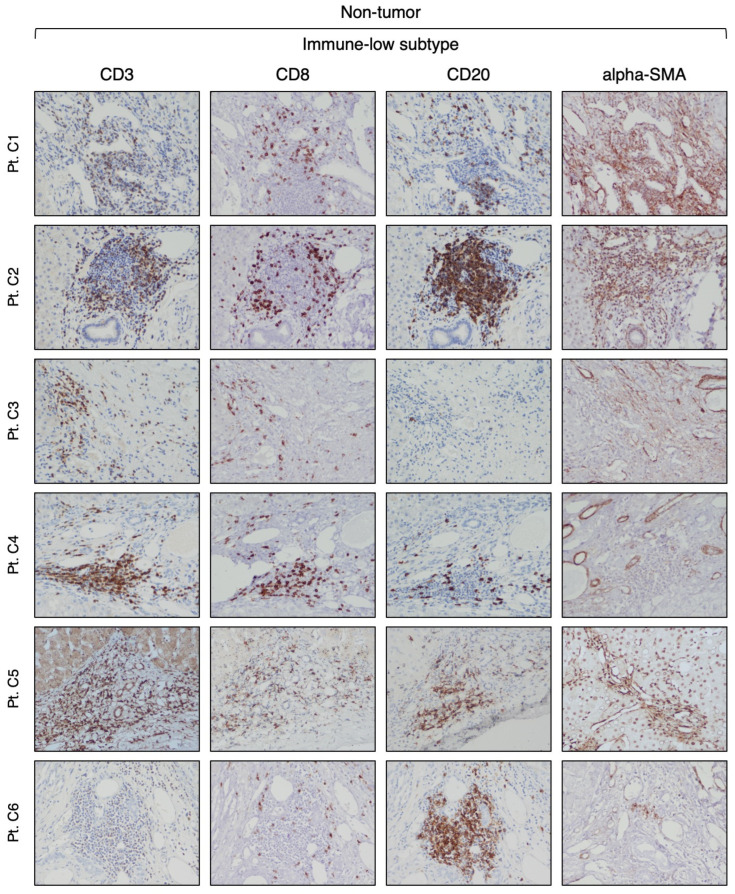
Expression of immune markers in the surrounding nontumorous tissue of the immune-low HCC subtype. The images illustrate the immunostaining with monoclonal antibodies against CD3 (T cells), CD8 (CD8 T cells), CD20 (B cells), and alpha-SMA (hepatic stellate cells) in paraffin liver sections from six HBV-HCC patients with the immune-low subtype taken at the time of liver transplantation from the most distant area from the center of the tumor (area D) (200× magnification).

**Figure 6 cancers-16-01370-f006:**
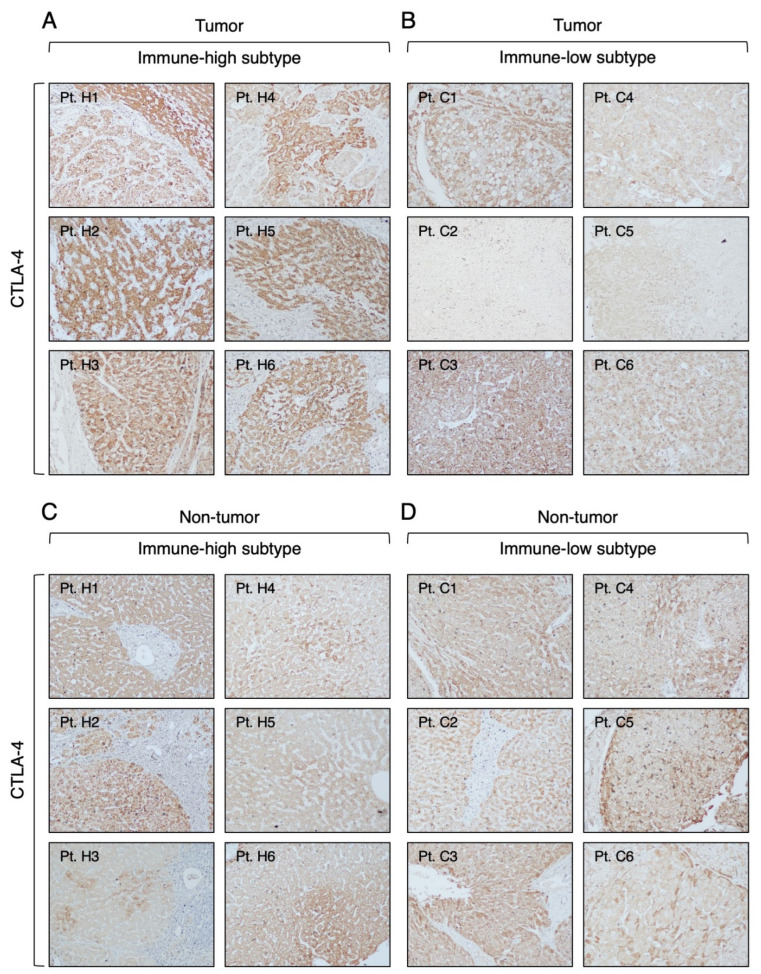
Expression of CTLA-4 molecule in the tumor and nontumorous tissue of immune-high and immune-low subtypes. The images illustrate the immunostaining with monoclonal antibody against CTLA-4 in paraffin liver sections taken at the time of liver transplantation from the center of the tumor (**A**,**B**) and the surrounding nontumorous tissue (**C**,**D**). The left two columns show the immune-high subtype, and the right two columns show the immune-low subtype (100× magnification).

**Figure 7 cancers-16-01370-f007:**
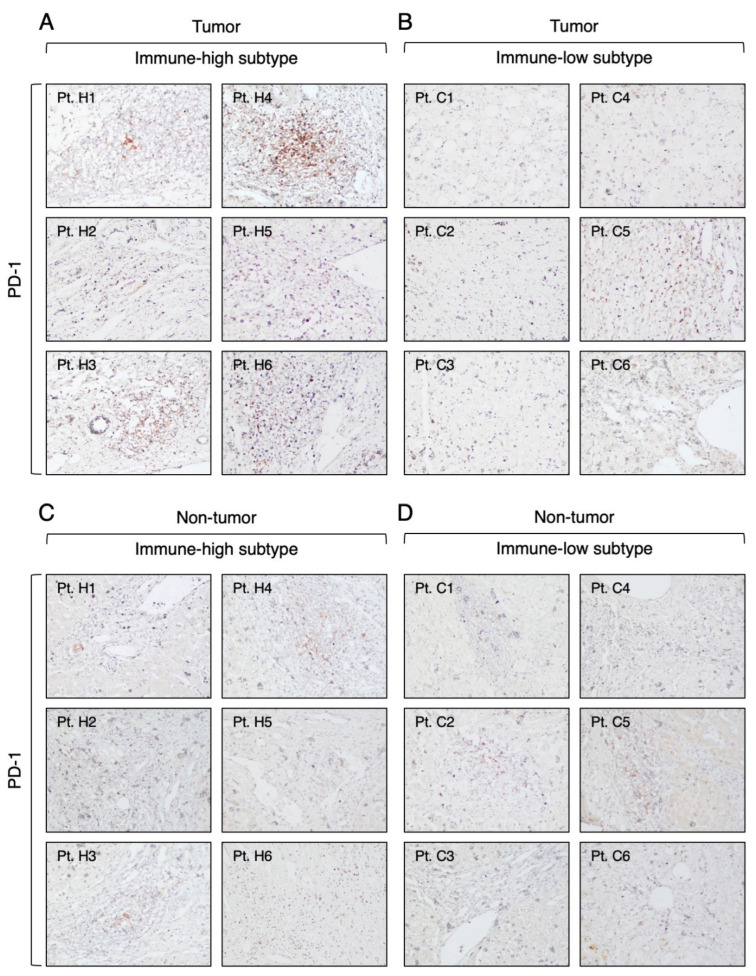
Expression of PD-1 molecule in the tumor and nontumorous tissue of immune-high and immune-low subtypes. The images illustrate the immunostaining with monoclonal antibody against PD-1 in paraffin liver sections taken at the time of liver transplantation from the center of the tumor (**A**,**B**) and the surrounding nontumorous tissue (**C**,**D**). The left two columns show the immune-high subtype, and the right two columns show the immune-low subtype (100× magnification).

**Figure 8 cancers-16-01370-f008:**
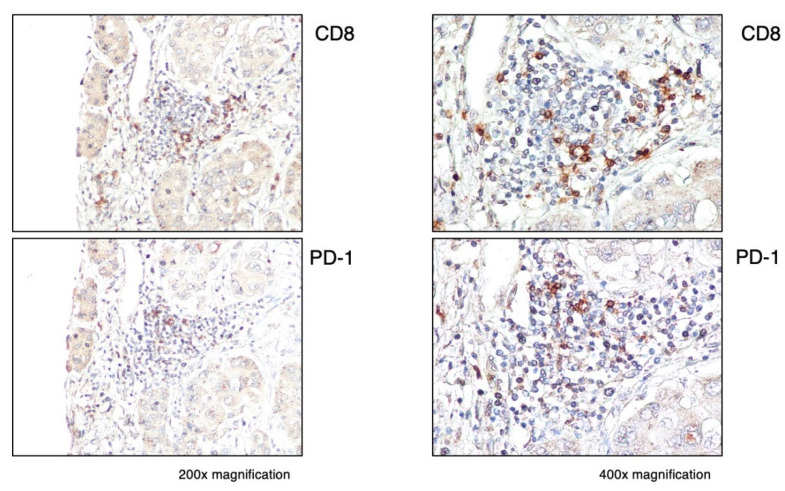
Expression of PD-1 within the tumor in a representative immune-high HCC case (Pt. H4). Liver specimens obtained at the time of liver transplantation from the center of the tumor were stained with monoclonal antibodies against CD8 (CD8 T cells) and PD-1. The images show the distribution of CD8 and PD-1 positive cells in the same cluster within the tumor (200× magnification in the left images and 400× magnification in the right images).

**Figure 9 cancers-16-01370-f009:**
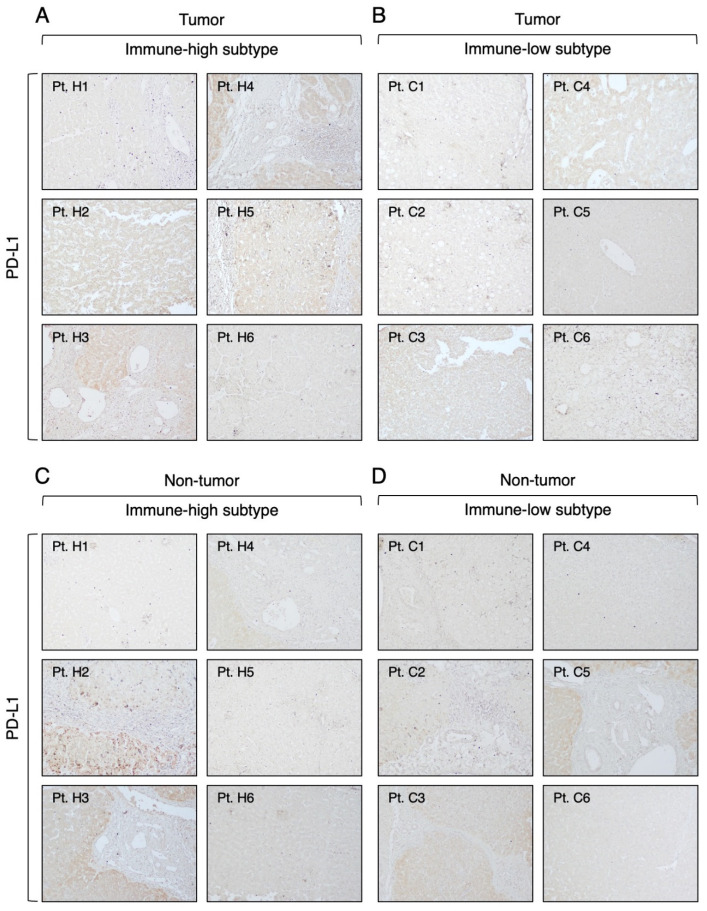
Expression of PD-L1 molecule in the tumor and nontumorous tissue of immune-high and immune-low subtypes. The images illustrate the immunostaining with monoclonal antibody against PD-L1 in paraffin liver sections taken at the time of liver transplantation from the center of the tumor (**A**,**B**) and the surrounding nontumorous tissue (**C**,**D**). The left two columns show the immune-high subtype, and the right two columns show the immune-low subtype (100× magnification).

**Figure 10 cancers-16-01370-f010:**
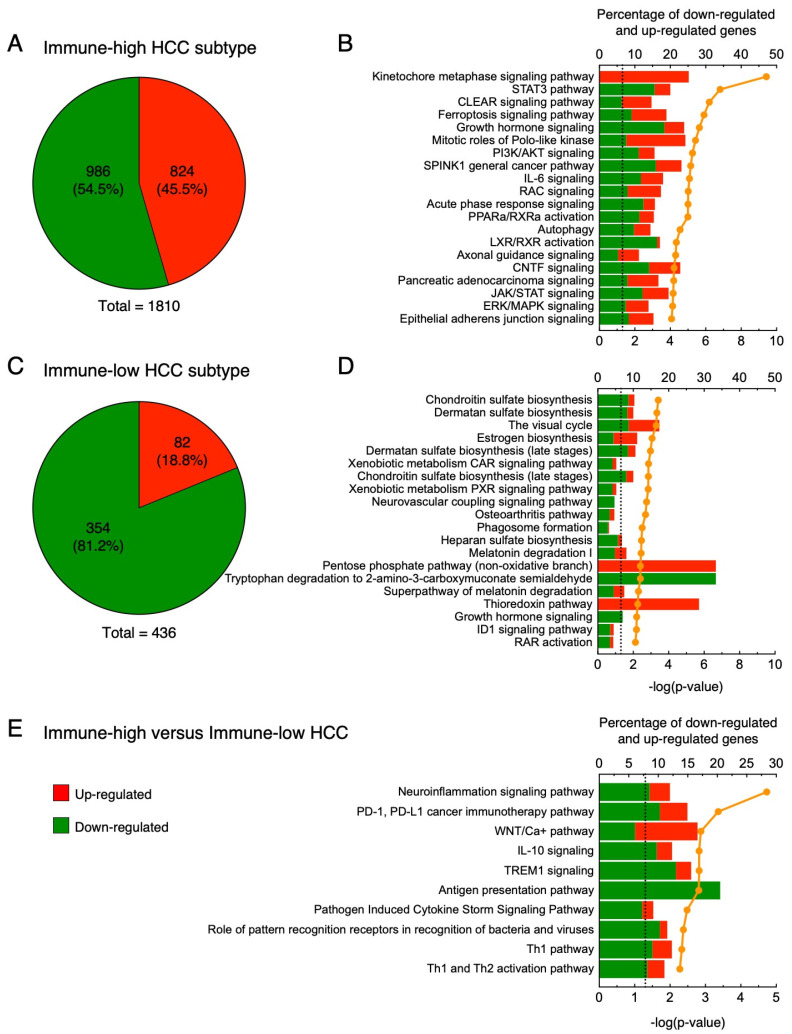
Transcriptomic analysis in the tumor versus nontumorous tissue of immune-high and immune-low subtypes: (**A**,**B**) immune-high HCC subtype; (**A**) pie charts showing the number of upregulated (red) and downregulated (green) genes; (**B**) twenty top-scored canonical pathways of differentially expressed genes obtained from whole liver tissue; (**C**,**D**) immune-low subtype; (**C**) pie charts showing the number of upregulated (red) and downregulated (green) genes; (**D**) twenty top-scored canonical pathways of differentially expressed genes obtained from whole liver tissue; (**E**) ten top-scored canonical pathways of differentially expressed genes from whole liver tissue samples obtained by comparing the transcription profile of the immune-high tumors versus the immune-low tumors. Columns (quoted on the top axes) represent the percent ratio between the number of genes present in the dataset and the total number of genes present in the database, for each pathway. The green and red portions of the columns indicate down-and upregulated genes, respectively. The orange line (quoted on the bottom axes) shows the statistical significance of each pathway, expressed as the negative log of the *p* value of Fisher’s exact test. The dotted lines point to the significance threshold corresponding to *p* = 0.05 on the log scale. Pathway analyses were obtained by Ingenuity Pathway Analysis (http://www.ingenuity.com, 21 November 2021).

**Figure 11 cancers-16-01370-f011:**
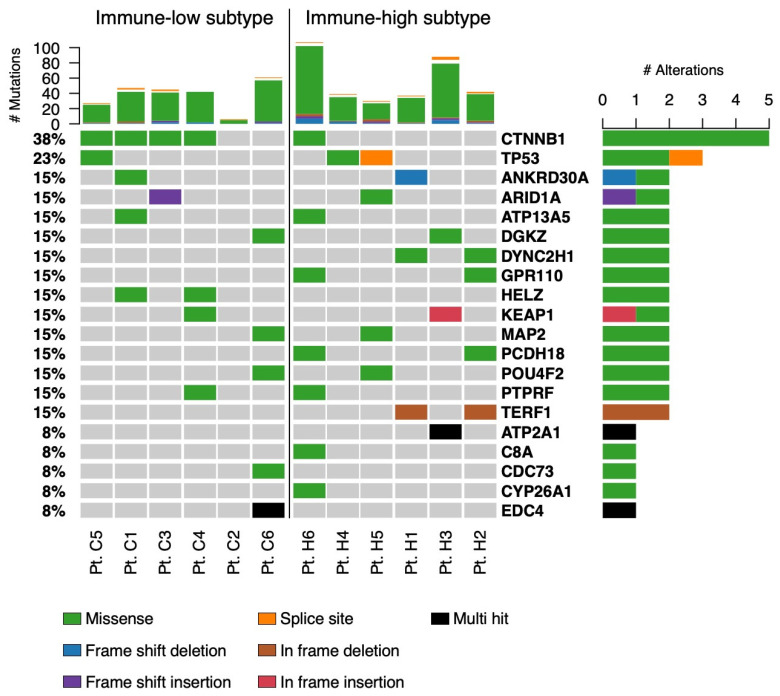
Mutational landscape in the two HCC immune subtypes. The top panel shows individual tumor mutation rates. The bottom panel shows the mutated genes identified in each patient. The mutation types are indicated in the legend at the bottom of the plot.

## Data Availability

The RNA-seq data presented in this study were previously deposited in the NCBI Sequence Read Archive (SRA), reference number PRJNA719288 (https://www.ncbi.nlm.nih.gov/sra/, 21 November 2021) [32]. The WES data have been deposited in the NCBI Sequence Read Archive (SRA) reference number PRJNA1085312 (https://www.ncbi.nlm.nih.gov/sra/, 21 November 2021).

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
