# Peer review of "Identification of Two Distinct Immune Subtypes in Hepatitis B Virus (HBV)-Associated Hepatocellular Carcinoma (HCC)"

_cancers, 2024, doi:10.3390/cancers16071370_

Round 1

Reviewer 1 Report

Comments and Suggestions for Authors

The manuscript presented the results of immunohistochemistry analysis from 12 cases of HBV-related HCC, after staining the markers of some types of immune cells, and the alpha-SMA that reflects stromal activation, and classical checkpoint molecules, PD-1, PD-L1 and CTLA-4. Meanwhile, the authors also presented the pathways of the genes differentially expressed between tumor and nontumorous tissue based on the data of the bulk RAN-seq data. The following results must be provided for improving the quality of the study. 

1. Mutations of some major driver genes of the two distinct tumor subtypes, immune-high and immune-low that was defined by the authors, such as the TP53, CTNN1, AXIN1..., as different mutations of these genes significantly affected the immune infiltration. Many previous studies have documented that the gene mutations of CTNN1, AXIN1, APC and some others resulted in the immune low phenotypes.  

2. Immunohistochemistry analysis was conducted on a series of sections. Sincer the manuscript would address the TME, no answers were provided regarding on the expression of immune checkpoint molecules on different types of immune cells or stromal cells. Colocalizations of the checkpoint molecules and cell markers of different types of cells must be provided.

3. Only 12 patients were included in the reports. The results were very primary. The finding must be validated in the other patients, at least the analysis based on some public data, such TCGA database. 

Author Response

Point-by-point response to reviewer #1

We thank this reviewer for his/her useful comments and suggestions, and we have modified the manuscript accordingly.

To improve the quality of our study, as suggested by this reviewer, we performed whole exome sequencing (WES) to study whether there was any difference in the mutational landscape between the two distinct immune subtypes. We did find that gene CTNNB was mutated, and the mutation of this gene was prevalent in the immune-low group. Thus, we have added new methods, new results, and a new figure (Figure 11), as well as additional references in the main text.

Because of the addition of the WES data, we have added Dr. Justin Lack as a coauthor.

Regarding the colocalization of the immune-checkpoints with different cell types, we would like to clarify that the expression of CTLA-4 and PD-L1 was detected on malignant and non-malignant hepatocytes, while PD-1 was detected only in immune cells infiltrating the liver specimens, as reported in the manuscript.

The markers used to identify the monocyte/macrophage lineage were chosen following the guidelines defined by Mantovani et al in 2015 [Mantovani et al Nat Rev Clin Oncol. 2017 Jul; 14(7): 399–416]. This reference has been added to the main text.

We agree with this reviewer that an additional comparison with gene expression data of HCC from deposited RNA-seq datasets (e.g., from The Cancer Genome Atlas Database, GEO, etc.) would add more confidence and we did try to obtain relevant data from the available database. However,

in the Cancer Genome Atlas Database, we did not find any relevant data, further highlighting the uniqueness of our study, in which paired liver specimens (tumor and surrounding nontumorous tissue) were obtained from well characterized patients, all infected by HBV. In particular, the available data considered HCC as a single group. As a consequence, the specific viral (HBV, HCV, or HDV) or nonviral etiology was missing. Likewise in GEO, a search performed using the keywords “HCC or hepatocellular carcinoma and RNA-sequencing”, yielded only very limited data: i) a small number of patients; ii) a different clinical scenario; and iii) a lack of direct comparison between paired liver specimens (tumor vs adjacent nontumorous tissue), as specified below.

  1. Sun et al. “A comprehensive genome-wide profiling comparison between HBV and HCV infected hepatocellular carcinoma”; BMC Med Genomics. 2019 Oct 28;12(1):147. TCGA-Liver Hepatocellular Carcinoma (HCC) cohort with publicly available data (https://www.cbioportal.org/study/summary?id=lihc_tcga_pan_can_atlas_2018;https://portal.gdc.cancer.gov/projects/TCGA-LIHC) was used for this study. The clinical characteristics of the patients were not well defined, and they did not compare the tumors with their surrounding non-tumorous tissues. They used as a control group para-cancerous tissues with HBV infection or without virus infection.

  1. Wheeler et al. “Comprehensive and Integrative Genomic Characterization of Hepatocellular Carcinoma” Cell. 2017 Jun 15;169(7):1327-1341.e23. The raw data are publicly available in the Genomic Data Commons (https://gdc.cancer.gov/). In this study, the etiology of HCC was not well defined, and they did not compare the tumors with their surrounding non-tumorous tissues.

  1. Sia et al. “Identification of an Immune-specific Class of Hepatocellular Carcinoma, Based on Molecular Features” Gastroenterology. 2017 Sep;153(3):812-826. The data are publicly available with GEO accession number GSE56588 and GSE63898. In this study, they examined the immune landscape of HCC, but for the RNA-seq data the etiology of HCC was not clearly defined, and the authors did not compare HBV- HCC with HCV- HCC. In addition, the authors did not compare the tumors with their surrounding non-tumorous tissues.

  1. Jin Y et al. “Comprehensive analysis of transcriptome profiles in hepatocellular carcinoma” J Transl Med. 2019 Aug 20;17(1):273. The data are available in Gene Expression Omnibus with series entry GSE105130. The aim of this study was different from ours.

Moreover, we further searched deposited data focusing on mRNA expression in HCC, but these analyses were performed using microarray platforms that cannot be directly compared with the more accurate RNA-seq analysis. Thus, despite our efforts, we were unable to identify relevant data from publicly available database. The limited number of samples, limited clinical information and different stage of HCC along with the lack of a direct comparison of paired liver specimens (tumor vs nontumorous tissue) have made it difficult to appropriately validate our results with those derived from HCC data publicly available.

However, we believe that the addition of WES, as suggested by the reviewers, has added a major value to our manuscript. The use of different molecular analyses (transcriptomics and WES) and the fact that all our patients were well characterized in terms of clinical, histopathological and virological features, and all were Caucasian seen at a single Italian Center, make our study unique and avoided epidemiological or demographic confounding factors.

Reviewer 2 Report

Comments and Suggestions for Authors

In this study, De Battista et al. characterized the tumor microenvironment of HBV-associated HCC tumors by imaging and transcriptomic approaches. They identified two distinct subtypes of HCC tumors depending on the infiltration of immune cells . The Immune high subtype is characterized by infiltration of B and T cells, stromal activation and high expression of CTLA4 while the low immune subtype dispalys lower lymphocyte infiltration. Furthermore, the transcriptomic profile associated to the high immune subtype strongly suggests that the infiltrated lymphocytes are not functional. This study is well-written and well-conducted and will be of interest of researchers in the field. 

The major comment I have is related to the markers used to identify macrophages and monocytes. Indeed, the authors used CD68 as monocyte/macrophage marker while it can also mark dendritic cells. Also, CD163 is considered as a M2-like macrophage marker in this study while in fact, CD163 appears to be specific to resident Kuppfer cells (Guilliams et al., Cell, 2022). Can the authors used more specific markers of infiltrating monocytes/macrophages to better characterize this population?

Also, the authors performed bulk RNA-seq to unveil the transcriptomic profiles of these two distinct subtypes. There are now several methods to deconvolute bulk RNA-seq data in order to appreciate the different cell types responsible for some responses. I think here deconvoluting the data anlysed here would complement the imaging data and bring more information. 

Finally, this classification has been obtained from HBV-associated HCC. Is it specific to this class of HCC or can HCC derived from the other aetiologies be classified similarly?

Author Response

Point-by-point response to reviewer #2

We appreciate the favorable comments of this reviewer as well as his/her useful comments and suggestions.

To improve the quality of our study, we performed whole exome sequencing (WES) to study whether there was any difference in the mutational landscape between the two distinct immune subtypes. We did find that gene CTNNB was mutated, and the mutation of this gene was prevalent in the immune-low group. Thus, we have added new methods, new results, and a new figure (Figure 11), as well as additional references in the main text.

Because of the addition of the WES data, we have added Dr. Justin Lack as a coauthor.

The markers used to identify the monocyte/macrophage lineage were chosen following the guidelines defined by Mantovani et al in 2015 [Mantovani et al Nat Rev Clin Oncol. 2017 Jul; 14(7): 399–416]. This reference has been added to the main text.

We appreciate the suggestion of this reviewer to use “methods to deconvolute bulk RNA-seq data in order to appreciate the different cell types responsible for some responses”. Unfortunately, we were unable to run this analysis. However, the expression of CTLA-4 and PD-L1 was detected on malignant and non-malignant hepatocytes, while PD-1 was detected only in immune cells infiltrating the liver specimens, as reported in the manuscript.

Regarding your question of whether our findings are specific to HBV HCC, our data show that the presence of an immune-high subtype is specific to this tumor and not to HCV HCC. In a previous paper published by our group in 2021 in Journal of Hepatocellular Carcinoma, entitled “Molecular Signature and Immune Landscape of HCV-Associated Hepatocellular Carcinoma (HCC): Differences and Similarities with HBV-HCC” De Battista et al. did not find any immune-high subtype in HCV HCC. This tumor was characterized by a dramatic downregulation of immune genes related to T-cell activation, a pattern consistent with the lack or the presence of very few infiltrating immune-cells (CD3, CD8 and CD20 positive cells) within the tumor in all HCV HCC cases examined. Thus, the finding of two distinct tumor subtypes, which we have defined as immune-high and immune-low in the present study, is new and unique to HBV HCC. These data bring new insights into HBV HCC pathogenesis and may be instrumental to identify patients who might benefit the most from immunotherapy.

We believe that the addition of WES, has added a major value to our manuscript. The use of different molecular analyses (transcriptomics and WES) and the fact that all our patients were well-characterized in terms of clinical, histopathological and virological features, and all were Caucasian seen at a single Italian Center, make our study unique and avoided epidemiological or demographic confounding factors.

Reviewer 3 Report

Comments and Suggestions for Authors

The author’s manuscript seems to provide very interesting and meaningful information on the tumor microenvironment of HBV-associated HCC, matching the results of two methods: histological analysis using immunostaining and transcriptomic analysis.

However, I would like to know about the two questions.

1) Regarding the sampling used in the RNA sequencing, please provide details on whether the samples were taken from the same area of the same field of views again used for immunostaining, or whether serial sectioned samples were used.

2) You have defined two subtypes of tumors, immune-high and immune-low, but have you examined the relationship between these subtypes and immunoregulatory cytokine levels?

Author Response

Point-by-point response to reviewer #3

We appreciate the favorable comments of this reviewer as well as his/her useful comments and suggestions.

To improve the quality of our study, we performed whole exome sequencing (WES) to study whether there was any difference in the mutational landscape between the two distinct immune subtypes. We did find that gene CTNNB was mutated, and the mutation of this gene was prevalent in the immune-low group. Thus, we have added new methods, new results, and a new figure (Figure 11), as well as additional references in the main text.

Because of the addition of the WES data, we have added Dr. Justin Lack as a coauthor.

Regarding the sampling used in the RNA sequencing, each sample was divided into two pieces: one was snap-frozen for RNA-sequencing and WES and the other was formalin-fixed and paraffin-embedded (FFPE) for pathological and immunohistochemistry analysis. Importantly, when FFPE sections obtained from the tumor or the perilesional area showed a mixed population (malignant and nonmalignant  hepatocytes), the corresponding liver specimen was excluded from immunohistochemical analysis and the frozen sample from RNA-sequencing and WES. The part regarding the method of WES has been added into the text.

We agreed with the reviewer that it is important to analyze whether the cytokine profile differed between the two distinct subtypes. We have attempted to address this question by analyzing the gene expression of the different cytokines, but we did not find any significant difference between the immune-high and the immune-low HCC subtypes.

Round 2

Reviewer 2 Report

Comments and Suggestions for Authors

The authors answered my concerns.